# A Gene from *Ganoderma lucidum* with Similarity to *nmrA* of Filamentous Ascomycetes Contributes to Regulating AreA

**DOI:** 10.3390/jof9050516

**Published:** 2023-04-26

**Authors:** He Liu, Jinjin Qiao, Jiaolei Shangguan, Xiaoyu Guo, Zhenzhen Xing, Xiaolin Zhou, Mingwen Zhao, Jing Zhu

**Affiliations:** Department of Microbiology, College of Life Sciences, Nanjing Agricultural University, Nanjing 210095, China; 2020116074@stu.njau.edu.cn (H.L.);

**Keywords:** nitrogen, AreA, *nmrA Basidiomyota*

## Abstract

Fungal AreA is a key nitrogen metabolism transcription factor in nitrogen metabolism repression (NMR). Studies have shown that there are different ways to regulate AreA activity in yeast and filamentous ascomycetes, but in *Basidiomycota*, how AreA is regulated is unknown. Here, a gene from *Ganoderma lucidum* with similarity to *nmrA* of filamentous ascomycetes was identified. The NmrA interacted with the C-terminal of AreA according to yeast two-hybrid assay. In order to determine the effect of NmrA on the AreA, 2 *nmrA* silenced strains of *G. lucidum*, with silencing efficiencies of 76% and 78%, were constructed using an RNA interference method. Silencing *nmrA* resulted in a decreased content of AreA. The content of AreA in *nmrA*i-3 and *nmrA*i-48 decreased by approximately 68% and 60%, respectively, compared with that in the WT in the ammonium condition. Under the nitrate culture condition, silencing *nmrA* resulted in a 40% decrease compared with the WT. Silencing *nmrA* also reduced the stability of the AreA protein. When the mycelia were treated with cycloheximide for 6 h, the AreA protein was almost undetectable in the *nmrA* silenced strains, while there was still approximately 80% of the AreA protein in the WT strains. In addition, under the nitrate culture, the content of AreA protein in the nuclei of the WT strains was significantly increased compared with that under the ammonium condition. However, when *nmrA* was silenced, the content of the AreA protein in the nuclei did not change compared with the WT. Compared with the WT, the expression of the glutamine synthetase gene in *nmrA*i-3 and *nmrA*i-48 strains increased by approximately 94% and 88%, respectively, under the ammonium condition, while the expression level of the nitrate reductase gene in *nmrA*i-3 and *nmrA*i-48 strains increased by approximately 100% and 93%, respectively, under the nitrate condition. Finally, silencing *nmrA* inhibited mycelial growth and increased ganoderic acid biosynthesis. Our findings are the first to reveal that a gene from *G. lucidum* with similarity to the *nmrA* of filamentous ascomycetes contributes to regulating AreA, which provides new insight into how AreA is regulated in *Basidiomycota*.

## 1. Introduction

Nitrogen is the raw material of protein, nucleic acid, and other biological macromolecules, and plays important roles in the growth and development of animals, plants, and microorganisms. How to improve nitrogen utilization and promote biological growth and metabolism have become hot research topics [1,2]. Fungi can use a variety of nitrogen sources, including inorganic and organic nitrogen sources. However, fungi preferentially utilize preferred nitrogen sources, such as ammonium salt and glutamine. When there is no preferred nitrogen source, fungi assimilate the non-preferred nitrogen sources, such as nitrate and proline. This regulatory strategy for nitrogen source utilization is called nitrogen metabolism repression (NMR) [3,4,5]. In fungi, NMR is mediated by GATA-type transcription factors. These are considered to be Gln3 and Gat1 in *Saccharomyces cerevisiae*, AreA in *Aspergillus nidulans* and *Ganoderma lucidum* [6,7,8], and NIT2 in *Neurospora crassa* [9].

The regulation of these GATA-type transcription factors on nitrogen metabolism has been reported in many fungi [3,10,11]. In addition, AreA has important impacts on the growth and development, virulence, and secondary metabolism of fungi [3,12,13]. However, the mechanism of nitrogen source regulation by AreA activity has mainly been studied in yeast and *Aspergillus*. In *S. cerevisiae*, under the preferred nitrogen condition, Gln3 and Gat1 were phosphorylated by interacting with the phosphorylated Ure2 and then located in the cytoplasm. Under the nitrogen starvation condition, Ure2 was dephosphorylated by Tap42-PP2A and dissociated from Gln3 and Gat1, which led to the transfer of Gln3 and Gat1 from the cytoplasm to the nucleus [14]. In *A. nidulans*, NmrA interacted with the C-terminal domain of the AreA protein that contains the zinc finger structure domain [15,16,17], and acted as a negative regulatory protein of AreA [18,19]. When the C-terminal of AreA or NmrA was deleted, some genes involved in nitrogen metabolism were overexpressed under the preferred nitrogen condition [18,20]. The NmrA protein in *A. nidulans* was gradually degraded when the external environment changed from preferred nitrogen to the nitrogen starvation condition [21]. However, the mechanism of how AreA is regulated in *Basidiomycota* has not been studied yet. According to the current research, the proteins involved in the regulation of AreA activity in yeast and some filamentous ascomycetes are different [22]; the regulation of AreA in *Basidiomycota* is worth studying further.

*G. lucidum* is an important medicinal basidiomycete, and has many active metabolites, such as ganoderic acid, polysaccharides, proteins, and steroids [23,24]. These active ingredients have important pharmacological functions [25]. The types or concentrations of nitrogen sources significantly affect the mycelial growth and the biosynthesis of active metabolites in *G. lucidum* [26,27]. Furthermore, AreA, a global transcription factor, played an important role in regulating the synthesis of secondary metabolites of *G. lucidum* [8,28]. However, there are few reports on how AreA responds to the changes in external nitrogen sources in *Basidiomycota*. Therefore, it is of great significance to study how AreA responds to nitrogen sources in *G. lucidum*.

In this study, we identified a homologous gene of the *Ganoderma nmrA*. The NmrA protein interacted with the C-terminal of AreA, and influenced the expression of genes involved in nitrogen metabolism, mycelial growth, and secondary metabolism. Further study showed that NmrA was required for the stability of the AreA protein, but it did not influence the nuclear localization of AreA. Our findings provide insight into how AreA is regulated in *Basidiomycota*.

## 2. Materials and Methods

### 2.1. The G. lucidum Strains and Growth Conditions

The *G. lucidum* strain (ACCC53264) was provided by the Agricultural Culture Collection of China (ACCC). Complete yeast medium (CYM) was used to culture the strains. The components of the CYM included 2% glucose, 1% maltose, 0.2% tryptone, 0.2% yeast extract, 0.4% KH_2_PO_4_, and 0.05% MgSO_4_ (pH 5.5). *Escherichia coli* DH5α (Sangon Biotech, Shanghai, China) was grown in Luria–Bertani (LB) media containing ampicillin (100 μg·mL^−1^) or kanamycin (50 μg·mL^−1^) for extracting the constructed plasmid. Different nitrogen media contained 2% glucose, 0.5% KH_2_PO_4_, 0.5% MgSO_4_, 15 mM (NH_4_)_2_SO_4_, or 30 mM NaNO_3_. The WT and *nmrA* silenced strains were cultured in media at 28 °C for 5 days. Finally, the colonies of the WT and *nmrA* silenced strains were photographed, and their diameters were measured. The WT and *nmrA* silenced strains were cultured in CYM liquid medium for 4 days, and then they were cultured under the ammonium or nitrate conditions for 3 days.

### 2.2. Analysis of NmrA and Ure2 and Construction of the nmrA Silenced Strains

The sequences of NmrA from *A. nidulans* (AAC39442.1), *Fusarium fujikuroi* (CAA75863.1), *N. crassa* OR74A (XP_961314.3), and Ure2 of *S. cerevisiae* (AAM93186.1) were obtained from NCBI. These sequences were used as queries in the local protein library of *G. lucidum* to obtain the homologous protein. SMART (http://smart.embl-heidelberg.de/, accessed on 28 March 2023) was used to analyze the conserved domain of Ure2 and NmrA. DNAMAN software was used for the homologous sequence alignment.

The total RNA of the samples was extracted with RNAisoTM Plus Reagent and a Hiscript^®^III RT SuperMix for real-time PCR (RT-qPCR) (Vazyme, Nanjing, China) was used to convert the RNA into cDNA. Using the paired primers listed in Table 1, with the cDNA of *G. lucidum* as the template, the silencing sequences of *nmrA* were obtained by PCR. The PCR products were digested using *Kpn*I and *Xba*I and then cloned into the previously constructed pAN7-*ura*3-dual-*hyg* vector [29], in which a glyceraldehyde-3-phosphate dehydrogenase promoter was used to express the antisense strand of *nmrA*. The constructed pAN7-*nmrA*-*hyg* plasmids were introduced into the protoplast of the wild-type (WT) strain by an electroporation method [30]. A CYM medium containing 100 μg·mL^−1^ hygromycin B (Sangon Biotech, Shanghai, China) was used to select the transformants, and the silencing efficiencies of transformants were verified by RT-qPCR with the primers listed in Table 1. The 18S RNA was used as the housekeeping gene. A ChamQ SYBR qPCR Master Mix (Vazyme, Nanjing, China) and the 2^–ΔΔCT^ method (Livak and Schmittgen 2001) were used to analyze all samples on a Mastercycler^®^ep realplex (Eppendorf, Hamburg, Germany). The *nmrA* silenced strains were named *nmrA*i strains, and the WT strain transformed with pAN7-*ura*3-dual-*hyg* plasmid was named the empty-vector control (CK).

### 2.3. Yeast Two-Hybrid Assay

The cDNA of *G. lucidum* was used as the template, and the primers in Table 1 were used to obtain the sequences of the full length of *areA*, the N- and C-terminal of *areA*, and *nmrA* by PCR. These sequences were ligated to the pGBKT7 bait vector or the pGADT7 prey vector. Yeast Y_2_H gold strains were co-transformed by pGBKT7-*areA* and pGADT7-*nmrA*. The Y_2_H gold strains were screened on synthetic dropout media (SD-leu/Trp/Ade/AbA/X-α-gal/15mM 3-AT) at 28 °C for 3–5 days, and the interactions were assessed on SD media (SD-leu/Trp /His/Ade/AbA/X-α-gal/5 mM 3-AT). Protein interactions were assessed based on the expression of different reporters under the control of GAL4-responsive promoters [31]. The yeast strains co-transformed with pGBKT7-53 and pGADT7-T were used as the positive control, and pGBKT7-Lam and pGADT7-T were used as the negative control. Protein interactions were assessed based on the production of blue colonies on the plates.

### 2.4. Protein Extraction

For the extraction of total protein, 0.1 g of fresh mycelia was ground into a powder with liquid nitrogen and added to 800 μL of extraction buffer, which contained 0.5 M Tris-HCl, 10% SDS, 20% glycerin, 1 M dithiothreitol, and 100 mM phenylmethyl sulfonyl fluoride. All samples were placed on ice for 10 min and then centrifuged at 10,000× *g* at 4 °C for 10 min. The supernatant was taken as the protein sample. The NE-PERTM Nuclear and Cytoplasmic Extraction Reagents (Thermo Scientific, Shanghai, China) were used to extract the nuclear and cytoplasmic proteins. In brief, 0.04 g of fresh mycelia was weighed and fully ground. According to the manufacturer’s instructions, an extraction buffer was added to the samples to extract the cytoplasmic and nuclear proteins, respectively.

### 2.5. Western Blot Analysis

Protein samples and 1 × SDS buffer (65 mM Tris-HCl, pH 6.8, 2% *w*/*v* SDS, 10% glycerol, 2.5% *v*/*v* β-mercaptoethanol, and 0.05% *w*/*v* bromophenol blue) were directly mixed and boiled at 100 °C for 10 min. The protein samples were separated on SDS-PAGE gels (12% polyacrylamide) and transferred to the polyvinylidene difluoride membranes using a Trans-Blot SD Cell and Systems (Bio-Rad, Hercules, CA, USA). After being blocked in 5% milk for 1 h at room temperature, the membranes were incubated with the primary antibody of β-tubulin (diluted 1: 1000) protein or AreA protein (diluted 1: 1000) overnight at 4 °C. The AreA protein was expressed and purified in our previously study [8]. The purified protein was sent to Genscript Biotech Corporation (Nanjing, China) to obtain the polyclonal antibody of AreA. The membranes were washed three times with TBST buffer (0.1% Tween-20 in 1 × TBS buffer) before being incubated with an appropriate secondary antibody. HRP goat anti-Mouse IgG antibody (diluted 1: 1000) and anti-Rabbit IgG antibody (diluted 1: 1000) were used as the secondary antibodies. The membranes were incubated with the secondary antibody for 1 h at room temperature, and then they were incubated using an ECL Western Blotting Detection Kit (Vazyme, Nanjing, China) and detected by a chemiluminescence detection system (Bio-Rad, Hercules, CA, USA).

### 2.6. Enzyme Activity Analysis

The fresh mycelia (0.5 g) were ground in liquid nitrogen and 2 mL of extraction buffer, which contained 50 mM Tris-HCl (pH 8.0), 2 mM MgSO_4_·7H_2_O, 4 mM DTT, and 400 mM sucrose, was added to the samples. All samples were centrifuged at 12,000× *g* at 4 °C for 15 min. The supernatant was collected for subsequent detection. The activity of glutamine synthetase (GS) was detected according to our previous methods [32]. The activity of nitrate reductase (NR) was detected by a Solarbio kit (Solarbio, Beijing, China). According to the guidelines of the kit, the reaction reagent was added to the supernatant, and finally, the absorbance at 340 nm was detected. The NR activity was calculated as an increase in the OD_340_ per 1 mg of protein after a 1 h reaction.

### 2.7. Quantification of Ganoderic Acid (GA)

The total GA was detected using the HPLC method according to the previously described method [33]. Briefly, dried mycelia (0.05 g) were weighed and fully ground and then added to 2 mL of a 10% (*w*/*v*) KOH-75% (*v*/*v*) ethanol solution. The samples were left for 2 h for saponification and the mixture was extracted 3 times with 2 mL hexane. Samples of the hexane layer were collected and vaporized under nitrogen gas until dry. The residue was dissolved in 0.5 mL of acetonitrile. An Agilent 1290 Infinity UPLC system (Agilent Technologies, Santa Clara, CA, USA) was used with an Agilent 1290 diode array detector and an Agilent Zorbax Eclipse Pluss C18 Rapid Resolution HD 18-Micron column (2.1 ×100 mm) (Agilent, Tokyo, Japan).

### 2.8. Statistical Analysis

The experiments was repeated three times. Each error bar represents the standard deviation (SD) of the mean of three replicates. Statistical analysis was performed using the one-way ANOVA provided by GraphPad Prism 8.0 software.

## 3. Results

### 3.1. A Homologous NmrA Protein Was Found in G. lucidum

AreA is the key transcription factor that functions in nitrogen metabolism, fungal growth, and secondary metabolism [8,34,35]. It was reported that NmrA in filamentous ascomycetes interacted with AreA, and Ure2 in *S. cerevisiae* interacted with Gln3 and Gat1 [22], but the AreA-interacting proteins in *Basidiomycota* are unknown. Here, the *ure2* (GenBank accession number: NP_014170.1) of *S. cerevisiae* and *nmrA* (GenBank accession number: XP_681437.1) of *A. nidulans* were used to determine the homologous genes according to the local genome bank of *G. lucidum*.

A suspected *nmrA* homologous gene (GenBank accession number: OP846035) was obtained in *G. lucidum*. The cDNA sequence of this *nmrA* gene had an open reading frame of 987 bp and encoded 329 amino acids with a deduced protein molecular weight of 36.74 kDa. The NmrA of *G lucidum* exhibited 60.03% amino acid identity with that of filamentous ascomycetes, including *F. fujikuroi* and *N. crassa* (Appendix A). The NmrA protein contained a Rossmann-type folded NAD^+^ and an NADP^+^ binding domain (Pfam05368, cd08947) (Figure 1A). This domain enables NmrA to distinguish between the oxidized and reduced forms of dinucleotides in *A. nidulans* [36]. The short chain dehydrogenase/reductase (SDR) family also contains a Rossmann folding domain. The difference between NmrA and SDR is that the proteins of SDR family have a Tyr-X-X-X-Lys structure, which is crucial for SDR activity [37]. However, the tyrosine in the Tyr-X-X-X-Lys structure is replaced by methionine, which leads to different functions between the NmrA and the SDR family [38]. We analyzed the sequences of NmrA in *G. lucidum* and found that the NmrA had the Met-X-X-X-Lys structure (Figure 1B). These results indicate that the homologous protein of NmrA was identified in *G. lucidum*.

A total of 4suspected homologous proteins of Ure2 (Gl23507, Gl23543, Gl24307, and Gl26599) were found in *G. lucidum* that had a conserved glutathione S-transferase (GST) domain (Appendix A) according to the Ure2 protein (the amino acids sequence accession number: AAM93186.1) in *S. cerevisiae*. The Ure2 in *S. cerevisiae* has a “flexible cap region” that determines its different function from glutathione S-transferase [39,40]. However, we found that the four suspected proteins of Ure2 in *G. lucidum* did not contain this special domain (Figure 1C), which indicates that these proteins of *G. lucidum* might not be homologous to the Ure2 of *S. cerevisiae*. In summary, the NmrA homologous protein in *G. lucidum* was used for studying further.

### 3.2. The NmrA Interacted with the C-Terminal of AreA According to a Yeast Two-Hybrid Assay

As a negative regulatory protein in nitrogen metabolism, NmrA interacted with AreA in *A. nidulans* and Nit2 (the homologous protein of AreA) in *N. crassa* [15,17]. Therefore, a yeast two-hybrid experiment was used to verify the interaction of AreA with NmrA in *G. lucidum*. The C-terminal of the AreA contained a zinc finger structure domain while the N-terminal contained a pfam DUF1752 domain. The full length of the *nmrA* was cloned into a pGADT7 prey vector. The sequences of *areA*-N and *areA*-C (Figure 2A) and the full length of *areA* were inserted into the pGBKT7 bait vector, respectively. These constructed vectors were then transformed into the yeast Y_2_H Gold strains. As shown in Figure 2B, yeast strains transformed with the pGADT7-*nmrA* and pGBKT7-*areA*-N grew on an SD-Trp/Leu medium. However, they did not grow on an SD-Trp/Leu/His/Ade medium. The above results showed that there was no interaction between AreA-N and NmrA. On the contrary, the yeast strains transformed with pGADT7-*nmrA* and pGBKT7-*areA*-C could grow on SD-Trp/Leu/His/Ade medium. The results of the yeast two-hybrid experiment showed that NmrA interacted with the C-terminal domain of AreA in *G. lucidum*.

### 3.3. Effect of NmrA on the Content of AreA Protein

In order to study the effect of NmrA on AreA, we first constructed the *nmrA* silenced strains using an RNA interference method. The silencing efficiencies of the strains were determined using RT-qPCR. Finally, four *nmrA* silenced strains, *nmrA*i-1, *nmrA*i-3, *nmrA*i-30, and *nmrA*i-48, were selected, with the expression reduced by approximately 74%, 79%, 77%, and 72%, respectively, compared with the WT (Figure 3A). Under the ammonium condition, the *nmrA* expression of *nmrA*i-1, *nmrA*i-3, *nmrA*i-30, and *nmrA*i-48 was decreased by approximately 71%, 68%, 76%, and 67%, respectively, compared with that of WT (Figure 3B). Under the nitrate condition, the *nmrA* expression of *nmrA*i-1, *nmrA*i-3, *nmrA*i-30, and *nmrA*i-48 was decreased by approximately 73%, 69%, 74%, and 77%, respectively, compared with that of WT (Figure 3C). *nmrA*i-3 and *nmrA*i-48 were randomly selected for the following experiments.

We examined the effect of NmrA on the expression of *areA* and the AreA protein content when mycelia were cultured in ammonium or nitrate conditions as sole nitrogen sources. As shown in Figure 3D, silencing of *nmrA* did not influence the expression of *areA* either in ammonium or nitrate. However, the intracellular AreA protein content was significantly reduced in the *nmrA* silenced strains compared with that in the WT (Figure 3E). Under the ammonium condition, the protein content of AreA in *nmrA*i-3 and *nmrA*i-48 was decreased by approximately 65% and 70%, respectively, compared with that of the WT (Figure 3F). While under the nitrate condition, the content of AreA in the *nmrA*i-3 and *nmrA*i-48 silenced strains was decreased by approximately 30% and 46%, respectively, compared with that of the WT (Figure 3F). These results indicate that silencing *nmrA* decreased the content of AreA.

### 3.4. Effect of NmrA on the Stability of the AreA Protein

In order to further study the effect of NmrA on the stability of the AreA protein, cycloheximide (CHX) was used to treat the mycelia cultured on CYM to detect the degradation of the AreA protein. CHX is an inhibitor of eukaryotic protein synthesis and is usually used to detect protein degradation [41,42]. After the WT and *nmrA* silenced strains were treated with CHX, the protein content of AreA was detected by Western blot. As shown in Figure 4, the protein content of AreA in the WT was decreased gradually with the prolongation of the CHX treatment time. At 6 h, approximately 80% of the AreA protein was still present, compared with the untreated strains. Compared with the change in AreA protein content in the WT, the AreA content in *nmrA* silenced strains decreased faster. After 6 h of treatment, the AreA protein content was nearly undetectable in the *nmrA* silenced strains. These results indicate that silencing *nmrA* reduced the stability of the AreA protein.

### 3.5. Effect of NmrA on the Subcellular Localization of AreA

As a key transcription factor responsible for regulating nitrogen metabolism, AreA enters the nucleus and regulates the expression of its downstream-related genes [20,43,44,45]. Therefore, the contents of AreA in the nucleus were detected under the ammonium or nitrate condition. As shown in Figure 5A,B, the content of AreA protein in the nuclei of the WT strain was increased 4.16-fold under the nitrate condition compared with that under the ammonium condition. The contents of the AreA protein in the nucleus of the *nmrA*i-3 and *nmrA*i-48 strains showed no significant changes compared with that of the WT, no matter the conditions (ammonium salt or nitrate) (Figure 5A,B). However, the content of the AreA protein in the cytoplasm of the WT, *nmrA*i-3, and *nmrA*i-48 strains was increased under the ammonium condition compared with those under the condition of nitrates (Figure 5C,D). These results suggest that although silencing of *nmrA* decreased the intracellular AreA content, it did not affect the AreA content of the nuclei.

### 3.6. Effect of NmrA on the Expression Levels of Glutamine Synthase Gene and Nitrate Reductase Gene

AreA can activate the expression of the glutamine synthase gene (*gs*) and the nitrate reductase gene (*nr*) by binding to their promoter regions in *G. lucidum* [32]. Therefore, we examined these gene expressions and enzyme activities. As shown in Figure 6A,B, compared with the WT, the expression of *gs* in *nmrA*i-3 and *nmrA*i-48 was increased by approximately 94% and 88%, respectively, and the enzyme activity was also increased by approximately 65% and 40%, respectively, under the ammonium condition (Figure 6B). However, compared with the WT, there were no significant changes in the expression level of *gs* or the enzyme activity of glutamine synthase under the nitrate condition. The expression and enzyme activity of *nr* showed different results from those of *gs*. Compared with the WT, the expression level or enzyme activity of nitrate reductase of *nmrA*i-3 and *nmrA*i-48 did not change significantly under the ammonium condition. However, under the nitrate condition, the expression of *nr* in *nmrA*i-3 and *nmrA*i-48 strains was increased by approximately 100% and 93%, respectively, and the activity of NR increased by approximately 64% and 45%, respectively (Figure 6C,D). These results indicate that silencing of *nmrA* influenced the expression levels of *gs* and *nr*.

### 3.7. Effect of NmrA on Mycelial Growth and Ganoderic Acid Biosynthesis

AreA has important impacts on fungal growth and secondary metabolism [3,8]. First, we investigated mycelial growth when *nmrA* was silenced. As shown in Figure 7A,B, compared with the WT, the growth of *nmrA*i-3 and *nmrA*i-48 were significantly inhibited under both the ammonium salt and nitrate conditions. Under the ammonium condition, the diameters of *nmrA*i-3 and *nmrA*i-48 were decreased by approximately 25% and 26%, respectively, compared with that in the WT. Under the nitrate condition, the diameters were decreased by approximately 33% and 38%, respectively, in the *nmrA*i-3 and *nmrA*i-48 strains. In addition, the biomasses of *nmrA*i-3 and *nmrA*i-48 were decreased by approximately 21% and 23%, respectively, compared with that of the WT under the ammonium condition. Under the nitrate condition, the biomasses of *nmrA*i-3 and *nmrA*i-48 were decreased by approximately 20% and 26%, respectively, compared with that of the WT (Figure 7C). The above results suggest that silencing of *nmrA* has a negative effect on mycelial growth.

The ganoderic acid (GA) contents in *nmrA*i-3 and *nmrA*i-48 were increased by approximately 25% and 31%, respectively, under the ammonium condition compared with the WT. However, the GA contents of *nmrA*i-3 and *nmrA*i-48 under the nitrate condition did not change significantly compared with the WT (Figure 7D). In addition, the expression levels of genes involved in GA biosynthesis, such as *hmgr* (3-hydroxy-3-methylglutaryl-CoA reductase), *sqs* (squalene synthase), and *osc* (lanosterol synthase), were also investigated (Appendix A). We found that the expression levels of *hmgr* and *osc* of *nmrA*i-3 and *nmrA*i-48 increased under the ammonium condition while the expression levels of *sqs* remained unchanged. Although the expression level of *hmgr* was decreased under the nitrate condition, the *sqs* expression level was increased, and the *osc* expression level remained unchanged. These results indicate that silencing of *nmrA* could influence GA biosynthesis.

## 4. Discussion

Fungi can use a wide range of nitrogen sources. The mechanism of how fungi control nitrogen utilization has been studied in yeast and filamentous ascomycetes. Fungal AreA is a key nitrogen metabolism transcription factor in NMR. Many studies have shown that there are different ways to regulate AreA activity in yeast and filamentous ascomycetes [22], but in *Basidiomycota*, how AreA is regulated is unknown. In this study, a homologous protein of NmrA was identified, and its interaction with AreA was shown in *G. lucidum*. In addition, NmrA was shown to be required for the stability of the AreA protein, mycelial growth, and secondary metabolism.

The NmrA in filamentous fungi is a negative regulatory protein in the NMR, and it plays important roles in fungal growth, development, and toxicity [18,19]. The growth of *A. nidulans* was significantly inhibited after the deletion of *nmrA* [18], and mycelial growth inhibition was also observed after the deletion of *nmrA* in *A. flavus* and *Metarhizium anisopliae* [19,46]. In addition, the deletion of *nmrA* in *A. flavus* resulted in increased conidia production and enhanced adaptation to the environment [46]. While the conidia yield and virulence decreased when *nmrA* was deleted in *M. anisopliae* [19], mutation of *nmrA* did not affect the virulence of *F. fujikuroi* [3,47]. Here, we found that the NmrA affect the mycelial growth and the ganoderic acid content of *G. lucidum*.

Fungal NmrA plays important roles in not only many physiological processes but also in nitrogen metabolism, such as nitrogen sources utilization. In *N. crassa*, glutamine reduced deaminase production. However, when *nmr-1* (the *nmrA* homologous gene) was mutated, the enzyme activity of deaminase was increased under the glutamine condition [48]. The NmrA of *G. lucidum* influenced the expression of genes involved in nitrogen metabolism, such as genes encoding glutamine synthetase and nitrate reductase. The expression of *areA* in WT was increased under the nitrate condition compared with that under the ammonium condition, and then the activated AreA regulated the expression of *gs* and *nr* [8,32]. Under the nitrate condition, silencing of *nmrA* led to a further increase in *nr* expression, which might be due to the reduced inhibitory effect of NmrA on AreA. However, the expression of *gs* showed the opposite result compared with the expression of *nr*. The *gs* expression did not change in the *nmrA* silenced strains under the nitrate condition. However, it increased in the *nmrA* silenced strains compared with that in the WT strains under the nitrate condition. The expressions of these genes are not only induced by AreA.The transcription factor NirA could act synergistically with AreA to mediate nitrate reductase enzyme activity in *Aspergillus* [49]. GCN4, a bZIP transcription factor, also has a pivotal role in the expression of nitrogen-regulated genes and nitrogen metabolism [50]. It interacted with Sko1 to form heterodimers to activate the expression of *areA* and *gs* in *G. lucidum* [51]. These studies not only demonstrate that other factors might participate in the regulation of the gene expression of *nr* and *gs*, but also reflect that the regulation of nitrogen utilization is a complex network.

Interaction of proteins, signaling, and post-translational modifications can control the functioning of transcription factors, including subcellular localization, protein stability, and transcriptional regulatory activity [52,53]. NmrA interacted with AreA to regulate the AreA activity in filamentous ascomycetes [15,16,17], but did not influence the entrance to the nucleus of AreA [20]. In *A. nidulans*, under the nitrogen starvation condition, the content of NmrA protein was decreased, and the inhibitory effect of NmrA on AreA activity was relieved [18,20,21]. In yeast, it was the Ure2 protein that interacted with Gln3 (a homologous protein of AreA) and regulated the transfer of Gln3 to the nuclei to activate gene expression [54]. Additionally, the Gln3 was phosphorylated under the preferred nitrogen condition and then maintained in the cytoplasm. Until now, the post-translational modification has been only found in yeast. In this study, we conducted a preliminary study on the subcellular localization, transcriptional regulatory activity, and protein stability of AreA in *G. lucidum*. It is NmrA, rather than Ure2, that interacted with AreA in *G. lucidum*. However, silencing of *nmrA* did not affect the content of AreA in the nuclei but affected the total content of AreA, especially the cytoplasmic AreA content in *G. lucidum*. The regulation of AreA may be different in different species, and the detailed mechanism needs to be further studied. Our studies show how the AreA activity is regulated in *Basidiomycota* and suggest clues for further research.

## Figures and Tables

**Figure 1 jof-09-00516-f001:**
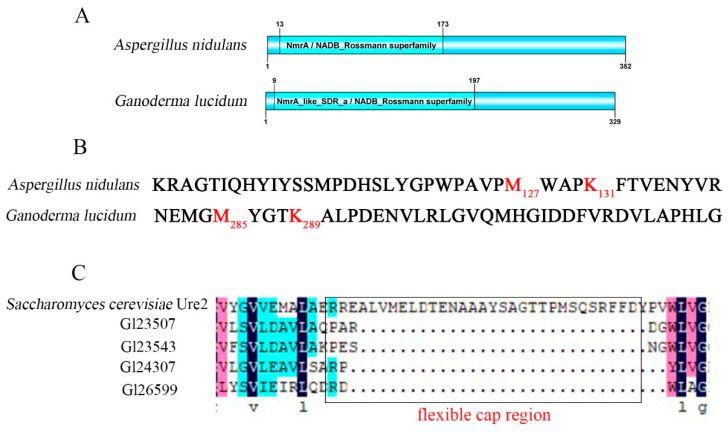
Bioinformatics analysis of NmrA in *G. lucidum* and *A. nidulans*, and Ure2 in *G. lucidum* and *S. cerevisiae*. (**A**) Conserved domain analysis of NmrA in *A. nidulans* and *G. lucidum*. (**B**) Determination of the Met-X-X-X-Lys structure in *A. nidulans* and *G. lucidum*. (**C**) Location analysis of the Ure2 “Flexible Cap Region” of *S. cerevisiae* and *G. lucidum*.

**Figure 2 jof-09-00516-f002:**
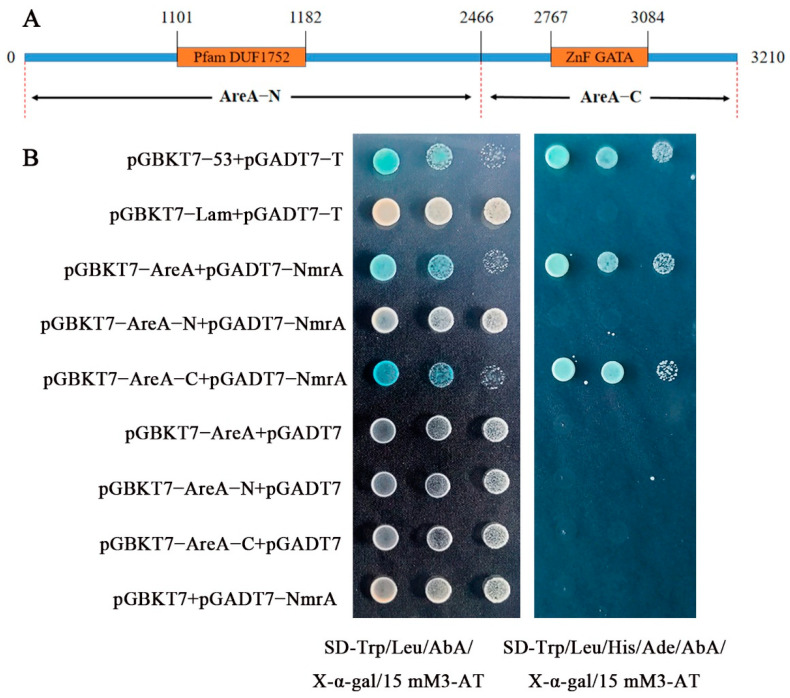
The yeast two-hybrid method was used to verify the interaction between AreA and NmrA. (**A**) The sequences of AreA were divided into two parts for the yeast two-hybrid assay. (**B**) pGBKT7-*areA*, pGBKT7-*areA*-N, or pGBKT7-*areA*-C was co-transformed with the pGADT7-*nmrA* into the Y_2_H strain. The Y_2_H strain containing the target carrier was cultured on an SD-Trp/Leu auxotrophic medium and an SD-Trp/Leu/His/Ade auxotrophic medium with 20 ng/mL of X-α-gal, 15 mM 3-AT, and 125 ng/mL AbA and then the media were incubated at 30 °C for 2 days to assess the protein–protein interactions. The strain with pGBKT7-53 and pGADT7-T was used as the positive control. The strain with pGBKT7-Lam and pGADT7-T was used as the negative control.

**Figure 3 jof-09-00516-f003:**
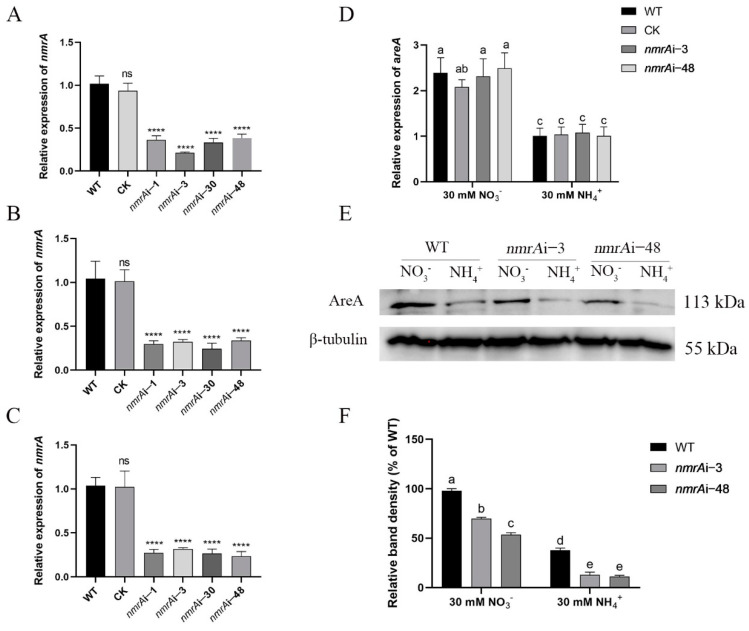
Effect of NmrA on AreA protein content. (**A**) Construction of the *nmrA* silenced strains. The strains were cultured on a CYM containing 100 μg/mL of hygromycin B at 28 °C for 5 days. The silencing efficiency of *nmrA* was identified by RT-qPCR. (**B**) The silencing efficiencies of *nmrA* of all strains were detected under the ammonium condition. (**C**) The silencing efficiencies of *nmrA* for all strains were detected under the nitrate condition. (**D**) The expression levels of *areA* in the WT and *nmrA* silenced strains were detected under the ammonium or nitrate conditions. (**E**) Western blot was used to detect the protein content of AreA in the WT and *nmrA* silenced strains. The WT and *nmrA* silenced strains were cultured in ammonium or nitrate at 28 °C for 5 days, and then the mycelia were collected for protein extraction. (**F**) The protein level was quantitatively analyzed using Image J v1.8.0 software. All data are presented as means ± SDs (n = 3). Asterisks indicate significant differences compared with the WT strain (one-way ANOVA, **** *p* < 0.0001, ns: no significance). Different letters indicate significant differences among groups (*p* < 0.05).

**Figure 4 jof-09-00516-f004:**
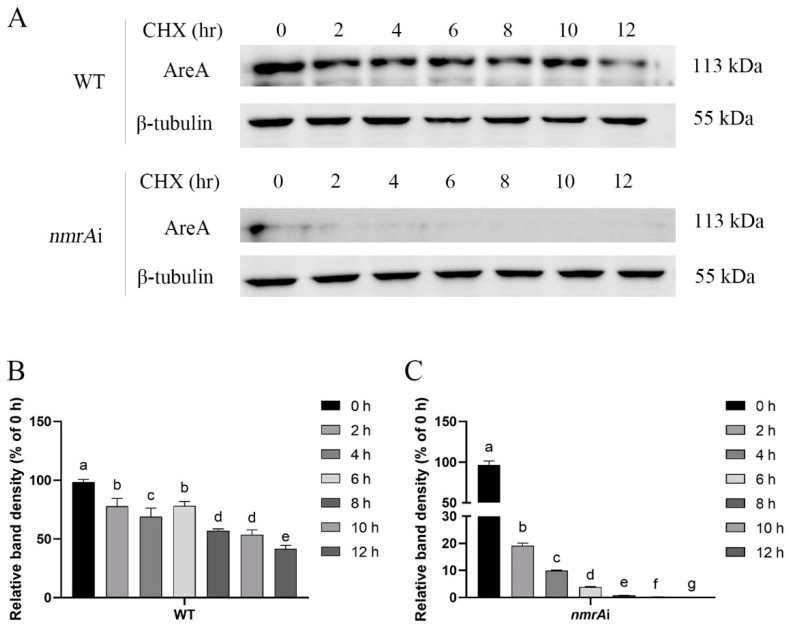
The effect of *nmrA* on the stability of the AreA protein. (**A**) The AreA protein levels of the WT and *nmrA* silenced strains were detected by Western blot. The WT and *nmrA* silenced strains were cultured on CYM and pretreated with cycloheximide (CHX) for 0 to 12 h. (**B**,**C**) The graphs show the relative band intensity of the AreA after different CHX treatments for the WT and *nmrA* silenced strains, respectively. Different letters indicate significant differences among groups (*p* < 0.05).

**Figure 5 jof-09-00516-f005:**
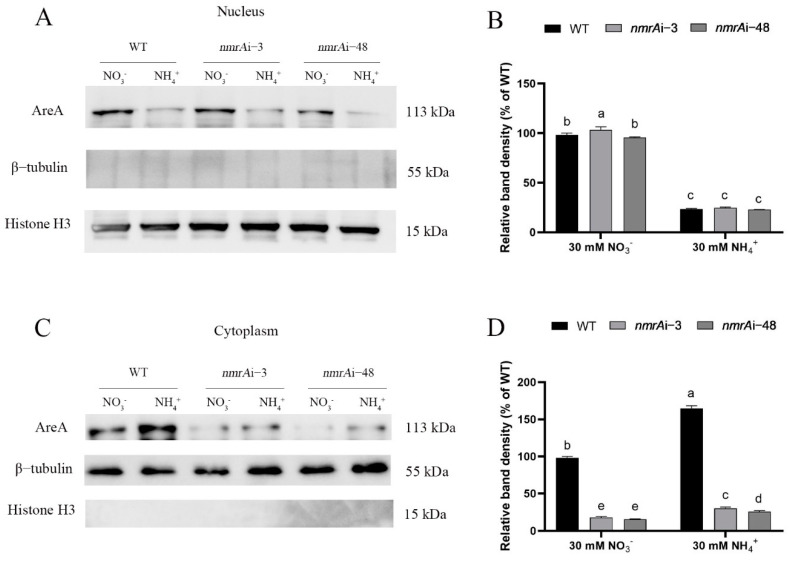
The effect of *nmrA* on the contents of AreA in the nuclei. (**A**) The contents of AreA in the nuclei of the WT and *nmrA* silenced strains under ammonium salt and nitrate conditions were detected by Western blot. (**B**) The protein content was quantified by Image J v1.8.0 software. The contents of AreA protein in the WT under the nitrate condition were used as the standard. (**C**) Western blot was used to detect the AreA protein contents in the cytoplasm of the WT and *nmrA* silenced strains under the ammonium or nitrate conditions. (**D**) The AreA protein content in (**C**) was quantified by Image J v1.8.0 software. The contents of AreA protein in the WT under the nitrate condition were taken as the standard. The Histone protein (H3) was used as an internal reference for the nuclear protein. The β-tubulin protein was used as an internal reference for the cytoplasm protein. All data are presented as means ± SDs (n = 3). Different letters indicate significant differences among groups (*p* < 0.05).

**Figure 6 jof-09-00516-f006:**
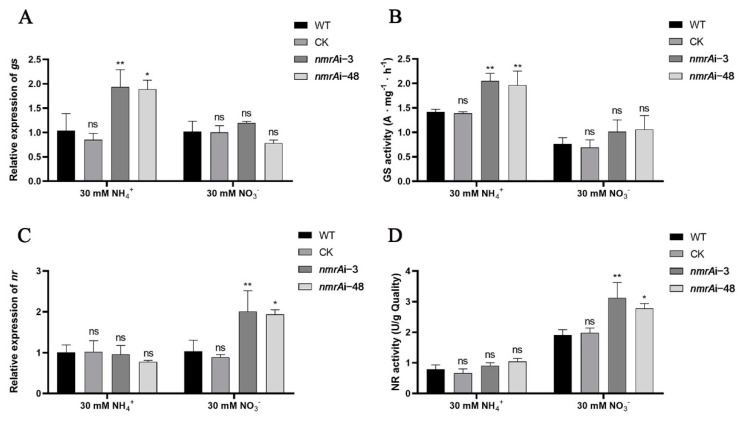
Expression levels and enzyme activities of glutamine synthase and nitrate reductase for the WT and *nmrA* silenced strains under the ammonium or nitrate conditions. (**A**) Relative expression of the *gs* gene in the WT, CK, and *nmrA* silenced strains under the ammonium or nitrate conditions. (**B**) The enzyme activity of GS for the WT, CK, and *nmrA* silenced strains under the ammonium and nitrate conditions. (**C**) Relative expression of the *nr* gene in the WT, CK, and *nmrA* silenced strains under the ammonium or nitrate conditions. (**D**) NR enzyme activity of the WT, CK, and *nmrA* silenced strains under the ammonium or nitrate conditions. All data are presented as means ± SDs (n = 3). Asterisks indicate significant differences compared with the WT strain (one-way ANOVA, * *p* < 0.05, ** *p* < 0.01, ns: no significance).

**Figure 7 jof-09-00516-f007:**
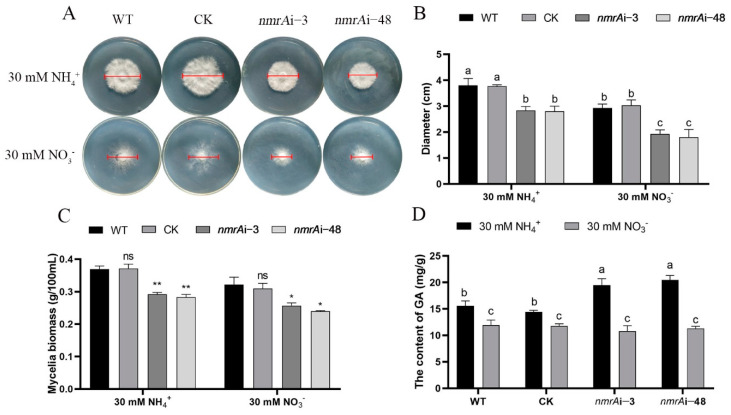
The effect of NmrA on mycelial growth and ganoderic acid biosynthesis. (**A**) The growth of the WT, CK, and *nmrA* silenced strains under the ammonium or nitrate conditions. All strains were photographed after 5 days of growth at 28 °C. (**B**) The diameters of the WT, CK, and *nmrA* silenced strains under the ammonium or nitrate conditions. (**C**) The mycelial biomasses of the WT, CK, and *nmrA* silenced strains under the ammonium or nitrate conditions. The WT, CK, and *nmrA* silenced strains were cultured in a CYM liquid medium for 3 days and then transferred to a liquid medium containing 30 mM of ammonium or 30 of mM nitrate for 5 days. Samples were then collected for detection. (**D**) The GA contents of the WT, CK, and *nmrA* silenced strains under the ammonium or nitrate conditions. All data are presented as means ± SDs (n = 3). Asterisks indicate significant differences compared with the WT strain (one-way ANOVA, * *p* < 0.05, ** *p* < 0.01, ns: no significance). Different letters indicate significant differences among groups (*p* < 0.05).

**Table 1 jof-09-00516-t001:** Paired primers designed for target gene manipulation in *Ganoderma lucidum*.

Primer	Sequence (5′ to 3′)	Description
pGBKT7-AreA-F	GGAATTCCATATGATGTTGCAACATACTCTC	Obtain full length of *areA* for yeast Y_2_H assay
pGBKT7-AreA-R	GGAATTCTTAAGCCCCGCCGCC	
pGADT7-NmrA-F	GGAATTCCATATGATGACGAAGCTCGTTGC	Obtain full length of *nmrA* for yeast Y_2_H assay
pGADT7-NmrA-R	GGAATTCTTACACGAGCGATAGGCCGAG	
pGBKT7-AreA-N-F	atggccatggaggccgaattcATGTTGCAACATACTCTC	Obtain N-terminal of *areA* for yeast Y_2_H assay
pGBKT7-AreA-N-R	tcgacggatccccgggaattcTTATGCGCCAGCACG	
pGBKT7-AreA-C-F	atggccatggaggccgaattcAGTGGCGCCCAGC	Obtain C-terminal of *areA* for yeast Y_2_H assay
pGBKT7-AreA-C-R	tcgacggatccccgggaattcTTAAGCCCCGCCGCC	
Hmgr-QRT-F	GTCATCCTCCTATGCCAAAC	Detect the *hmgr* expression
Hmgr-QRT-R	TGAACTGTGCGAAAGG	
Sqs-QRT-F	CTGCTTATTCTACCTGGTGCTACG	Detect the *hmgr* expression
Sqs-QRT-R	GGCTTCACGGCGAGTTTGT	
Osc-QRT-F	AGGGAGAACCCGAAGCATT	Detect the *hmgr* expression
Osc-QRT-R	CGTCCACAGCGTCGCATAAC	
GS-QRT-F	ACCAACTTCCGCCACCAT	Detect the *gs* expression
GS-QRT-R	AAGACCTTGCCAGCACCAG	
NR-QRT-F	AAGACGACCAACTCC	Detect the *nr* expression
NR-QRT-R	GCCAAGTGCCATAA	
18s-QRT-F	TATCGAGTTCTGACTGGGTTGT	Detect the 18s expression
18s-QRT-R	ATCCGTTGCTGAAAGTTGTAT	
NmrA-QRT-F	GCATATCTCCGGTGGTCGTT	Detect the *nmrA* expression
NmrA-QRT-R	GGTATTTGCCTGGACAACGC	
NmrAi-F	gcgcacaggcggagaactagtTCGTGCTCCTCTCGTTTG	Obtain the silencing fragment of the *nmrA* gene
NmrAi-R	actcttcatccccctggtaccCACGAGCGATAGGCCGAG	

## Data Availability

Not applicable.

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
