# Peer review of "A Gene from Ganoderma lucidum with Similarity to nmrA of Filamentous Ascomycetes Contributes to Regulating AreA"

_jof, 2023, doi:10.3390/jof9050516_

Round 1

Reviewer 1 Report (Previous Reviewer 2)

Liu and co-workers present the resubmission of their work on the identification of the possible homologue of NmrA in the basidiomycete Ganoderma lucidum.

Most of the previous concerns raised by this reviewer have been addressed and the manuscript is generally improved. It is now important to improve the use of English. Reading is difficult in some sections and reduces the understanding of the authors' message about their work and probably the interest of future readers. Experimental work is correct and conclusions are limited to findings. 

Author Response

Response: Thank you for your suggestions.

We have carefully revised the manuscript, especially the Introduction, Results and Discussion sections. In addition, combined with the suggestions of Reviewer 2 and Reviewer 3, the redundant parts of the article have been deleted.

We appreciate for your warm work earnestly, and hope that the correction will meet with approval.

Once again, thank you very much for your comments and suggestions.

Kind regards,

Jing Zhu

Reviewer 2 Report (Previous Reviewer 1)

This manuscript has been appropriately revised in several points. However, there are still some issues.

1. Although the experimental method has not changed, the results in Figure 4 differ from the previous manuscript. It is difficult to determine which result is more reliable. In addition, what nitrogen source was used in this experiment?

2. In Fig. 3E, there is no label for the strains.

3. Page 5, lines 196 and 218. The authors note the different GenBank accession number of ure2. Why?

4. Discussion section is not organized. For instance, there is a duplicate explanation about ure2. Overall rewrite is recommended.

5. All genus names must be indicated by abbreviation after the second time.

Author Response

This manuscript has been appropriately revised in several points. However, there are still some issues.

Response: Thank you for your suggestions. The point-to-point replies are listed below.

  1. Although the experimental method has not changed, the results in Figure 4 differ from the previous manuscript. It is difficult to determine which result is more reliable. In addition, what nitrogen source was used in this experiment?

Response: In the first version of manuscript, the Reviewer 1 pointed out that “AreA protein in nmrA silencing strain is suddenly disappeared 10 hours after CHX treatment. In the original image of fig. 4, the other proteins are also suddenly disappeared together with AreA. This is unnatural result.” Therefore, we have re-examined the stability of AreA protein in nmrA silenced strains for three times in the latest version, as shown in Fig. 4 and the original images. It was found that AreA protein decreased with the extension of treatment time, while the other non-specific proteins still existed. Therefore, these results were reliable.

In addition, in the result 3, the total AreA content decreased regardless of the nitrogen source used. Therefore, this experiment is carried out under the condition of CYM, which contains complete nitrogen sources. We have added the related description in the revised manuscript.

  1. In Fig. 3E, there is no label for the strains.

Response: We have added the labels for the strains the new Fig. 3E. Additionally, we have adjusted the position of images A-F to make the Figure 3 look more suitable.

  1. Page 5, lines 196 and 218. The authors note the different GenBank accession number of ure2. Why?

Response: Sorry for the unclear description. We have confirmed these two accession numbers and revised these two descriptions. In line 196, the gene of ure2 was used, and the gene sequence accession number was NP_014170.1. However, in line 218, the amino acids of Ure2 were used, and the amino acid sequence accession number is AAM93186.1.

  1. Discussion section is not organized. For instance, there is a duplicate explanation about ure2. Overall rewrite is recommended.

Response: Thank you for your suggestions. We have reorganized the discussion section of the manuscript.

  1. All genus names must be indicated by abbreviation after the second time.

Response: Thank you for your comments. We have checked and revised throughout the manuscript.

We appreciate for your warm work earnestly, and hope that the correction will meet with approval.

Once again, thank you very much for your comments and suggestions.

Kind regards,

Jing Zhu

Reviewer 3 Report (New Reviewer)

The authors show that the cytoplasmic level of AreA is regulated by the physical interaction of NmrA with AreA.

My main concern is that the results of the AreA functions examined, which were unexpected, were not interpreted. Biological insights into these results would greatly improve the manuscript.

E.g. Authors said that according to the literature, expression of gs depends on AreA. It would be useful to get an explanation from the Authors for the following discrepancy paradox result: the AreA protein level in the nuclei of nmrAi strains does not differ from the AreA level in the nuclei of WT under ammonium condition, but the expression of the AreA-regulated gs gene shows a significant increase in nmrAi strains. How this could be explained?

Furthermore, some parts of the Introduction and Results are redundant with the text of the Discussion. All redundant texts should be eliminated (detailed below). Additionally, the English usage of some parts of the manuscript should be improved to make the text more readable.

Minor concerns:

line 14: according to

lines 22-24: needs rewriting

line 35: delete the word biological from “biological protein”, as there is no such protein that is not biological.

line 42: Rephrase the sentence. Please, be more specific!

“This utilization of nitrogen sources is called nitrogen metabolism repression (NMR) [3-5].”

line 44: These are considered....

lines 50-80: this part of the text needs extensive rewriting due to poor usage of English

lines 85-87: need to be rephrased

“Our findings are the first to explain how AreA is regulated in G. lucidum, and they can an lay a foundation for the regulation of transcription factor AreA in Basidiomycota.”

line 196: S. cerevisiae

line 203: “The NmrA protein contained a Rossmann-type folded NAD+ and an NADP+ binding domain, which caused NmrA to discriminate between oxidized and reduced dinucleotides (Fig. 1A)..”

Based on the sentence above, a reader would have a false impression that the authors have performed experiments to prove the role of the Rossmann-fold domain. According to my interpretation, this is not true, hence this sentence needs to be rephrased, to better convey the Author’s message.

lines 204-215: poor usage of English

line 213: NmrA

line 236: The number of the pfam domain is missing. Please provide this essential information.

line 240: transformed

line 244: transformed

line 273: “Under the condition of ammonium salt”

Poor English phrasing.

lines 290-305: Make the symbol of P value uniform throughout the text.

In Panels D and F of Figure 3 the a, ab and c marks are not explained in the figure legend. If they do not denote P values, then explain these a, ab and c marks. If they denote P values, use the asterisk marks as in Panels A, B and C.

lines 292-305: Strain CK is not explained.

lines 318-324: marks a-g on Panels B and C of Figure 4 are not explained, please do so.

line 334: showed no significant changes

lines 331-335: Please rephrase these sentences, as right now the text is overcomplicated and does not completely reflect the experimental results presented in Figure 5 Panels A and B. Additionally, I noted that CK strain is not presented in the referred figure.

E.g.: As shown in Fig. 5A-B, the nmrAi strains showed a wt-like 4.16-fold increase of AreA content in the nuclei under the nitrate condition compared to the ammonium condition.

lanes 335-341: Simplify these sentences, as they are overcomplicated.

lanes 341-354: Marks a-e on Panels B and D of Figure 5 are not explained, please do so.

lane 359: “…and nr and enzyme activity…”  Delete the second and.

lanes 373-381: Strain CK is not explained.

lane 384: First, we investigated …

lanes 406-419: marks a-c on Panels B and D of Figure 5 are not explained, please do so.

lane 427: NmrA

lane 435: “… a nitrogen deficiency…”

Nitrogen-deficiency is vague. Please provide enough details for the reader and specify the exact condition properly. Do the authors mean nitrogen starvation?

lane 435: “…under normal nitrogen conditions…”

The same is true for this part, please provide enough details, what does normal nitrogen conditions mean? I assume when ammoniumor other preferred N-sources are available.

Lane 438: … nmrA was deleted

lane 439: “affect its virulence in F. fujikuroi

This is again an improper usage of English. The virulence of what?

I propose: … affects the virulence of F. fujikuroi

lane 452-454: Rephrase the sentence.

lane 463: separate is not the proper word in this context.

I propose:   …the dephosphorylation of Ure2 resulting in the dissociation of the Gln3-Ure2 complex…

lane 465: Delete the “For the regulation of the gs and nr gene,” from the sentence.

lanes 467-469: “However, the regulation of these genes does not always depend on NmrA because nmrA silencing did not always inhibit their expression under the nitrate conditions.

Please, be more specific. Avoid the phrase “does not always…”.

lane 470: What is NR in Aspergillus? Does NR refer to the nitrate reductase enzyme? There is no such historical abbreviation in Aspergilli, therefore, please avoid the usage of this abbreviation, or make it clear what does it mean.

lane 475: Basidiomycota

lanes 476-478: These sentences are redundant.

lanes 479-488: These sentences are redundant with the sentences in the Results section.

lanes 488-490: Redundant sentence.

lanes 490-499: These sentences are redundant with the Introduction.

Author Response

The authors show that the cytoplasmic level of AreA is regulated by the physical interaction of NmrA with AreA.

My main concern is that the results of the AreA functions examined, which were unexpected, were not interpreted. Biological insights into these results would greatly improve the manuscript.

E.g. Authors said that according to the literature, expression of gs depends on AreA. It would be useful to get an explanation from the Authors for the following discrepancy paradox result: the AreA protein level in the nuclei of nmrAi strains does not differ from the AreA level in the nuclei of WT under ammonium condition, but the expression of the AreA-regulated gs gene shows a significant increase in nmrAi strains. How this could be explained?

Response: Thank you for your comments. We have added the related discussion in line 426-442.

The NmrA of G. lucidum influenced the expression of genes involved in nitrogen metabolism, such as genes encoding glutamine synthetase and nitrate reductase. The expression of areA of WT increased under the nitrate condition compared with that under the ammonia condition, and then activated AreA regulated the expression of gs and nr. Under the nitrate condition, silencing nmrA led to a further increase in nr expression, which might be due to the reduced inhibitory effect of NmrA on AreA. However, the expression of gs showed the opposite results compared to the expression of nr. The gs expression did not change in the nmrA silenced strains under the nitrate condition. However, it increased in the nmrA silenced strains compared with that in the WT strains under the nitrate condition. The expression of these genes did not only induce by AreA. The transcription factors NirA could act synergistically with AreA to mediate nitrate reductase enzyme activity in Aspergillus. GCN4, a bZIP transcription factor, also has a pivotal role in genes expression and nitrogen metabolism. It interacted with sko1 to form heterodimers to activate the expression of areA in G. lucidum. These demonstrated the other factors might participate in regulating the expression of nr and gs, and also reflected that the regulation of nitrogen utilization is a complex network.

Furthermore, some parts of the Introduction and Results are redundant with the text of the Discussion. All redundant texts should be eliminated (detailed below). Additionally, the English usage of some parts of the manuscript should be improved to make the text more readable.

Response: Thank you for your comments. We have revised the parts of the Introduction, Results and Discussion. Additionally, combined with the reviewer 2’ suggestion, we have reorganized the discussion section.

Minor concerns:

line 14: according to

Response: We have revised accordingly.

lines 22-24: needs rewriting

Response: We have rewritten this sentence.

line 35: delete the word biological from “biological protein”, as there is no such protein that is not biological.

Response: We have revised accordingly.

line 42: Rephrase the sentence. Please, be more specific!

“This utilization of nitrogen sources is called nitrogen metabolism repression (NMR) [3-5].”

Response: We have rewritten this sentence.

line 44: These are considered....

Response: We have revised accordingly.

lines 50-80: this part of the text needs extensive rewriting due to poor usage of English

Response: We have rewritten these two paragraphs.

lines 85-87: need to be rephrased

“Our findings are the first to explain how AreA is regulated in G. lucidum, and they can an lay a foundation for the regulation of transcription factor AreA in Basidiomycota.”

Response: We have rewritten this sentence. Our findings provide an insight into how AreA is regulated in Basidiomycota.

line 196: S. cerevisiae

Response: We have revised the manuscript.

line 203: “The NmrA protein contained a Rossmann-type folded NAD+ and an NADP+ binding domain, which caused NmrA to discriminate between oxidized and reduced dinucleotides (Fig. 1A)..”

Based on the sentence above, a reader would have a false impression that the authors have performed experiments to prove the role of the Rossmann-fold domain. According to my interpretation, this is not true, hence this sentence needs to be rephrased, to better convey the Author’s message.

Response: Thank you for your suggestion. This domain enables NmrA to distinguish between the oxidized and reduced forms of dinucleotides in A. nidulans. We have re-written this sentence to avoid a false impression that we have proved its role.

lines 204-215: poor usage of English

Response: We have revised the manuscript.

line 213: NmrA

Response: We have revised accordingly.

line 236: The number of the pfam domain is missing. Please provide this essential information.

Response: We have added the number in the revised manuscript.

line 240: transformed

Response: We have revised accordingly.

line 244: transformed

Response: We have revised accordingly.

line 273:“Under the condition of ammonium salt”

Poor English phrasing.

Response: We have revised the manuscript.

lines 290-305: Make the symbol of P value uniform throughout the text.

In Panels D and F of Figure 3 the a, ab and c marks are not explained in the figure legend. If they do not denote P values, then explain these a, ab and c marks. If they denote P values, use the asterisk marks as in Panels A, B and C.

Response: Thank you for your comments. We have added the related description in the figure legend. Different letters indicate significant differences among groups (P < 0.05).

lines 292-305: Strain CK is not explained.

Response: Thank you for your comments. We have added the explanation of CK in the material method (page 119).

lines 318-324: marks a-g on Panels B and C of Figure 4 are not explained, please do so.

Response: Thank you for your comments. We have added the explanation of different letters.

line 334: showed no significant changes

Response: We have revised the manuscript.

lines 331-335: Please rephrase these sentences, as right now the text is overcomplicated and does not completely reflect the experimental results presented in Figure 5 Panels A and B. Additionally, I noted that CK strain is not presented in the referred figure.

E.g.: As shown in Fig. 5A-B, the nmrAi strains showed a wt-like 4.16-fold increase of AreA content in the nuclei under the nitrate condition compared to the ammonium condition.

Response: Thank you for your suggestions. We have re-written these sentences.

lanes 335-341: Simplify these sentences, as they are overcomplicated.

Response: Thank you for your suggestions. We have re-written these sentences.

lanes 341-354: Marks a-e on Panels B and D of Figure 5 are not explained, please do so.

Response: Thank you for your comments. We have added the explanation in the figure legend.

lane 359: “…and nr and enzyme activity…”  Delete the second and.

Response: We have revised accordingly.

lanes 373-381: Strain CK is not explained.

Response: We have explained CK in the material method line 119.

lane 384: First, we investigated …

Response: We have revised the manuscript.

lanes 406-419: marks a-c on Panels B and D of Figure 5 are not explained, please do so.

Response: We have added the explanation in the figure legend.

lane 427: NmrA

Response: We have revised accordingly.

lane 435: “… a nitrogen deficiency…”

nitrogen limitation

Nitrogen-deficiency is vague. Please provide enough details for the reader and specify the exact condition properly. Do the authors mean nitrogen starvation?

Response: Thank you for your comments. We have revised the manuscript. The nitrogen deficiency condition refers to the nitrogen starvation condition.

lane 435: “…under normal nitrogen conditions…”

The same is true for this part, please provide enough details, what does normal nitrogen conditions mean? I assume when ammonium or other preferred N-sources are available.

normal nitrogen conditions

Response: Thank you for your comments. We have revised the manuscript. The normal nitrogen condition represents the preferred nitrogen condition.

Lane 438: … nmrA was deleted…

Response: We have revised the manuscript.

lane 439: “affect its virulence in F. fujikuroi”

This is again an improper usage of English. The virulence of what?

I propose: … affects the virulence of F. fujikuroi

Response: We have revised accordingly.

lane 452-454: Rephrase the sentence.

Response: We have revised the manuscript.

lane 463: separate is not the proper word in this context.

I propose:   …the dephosphorylation of Ure2 resulting in the dissociation of the Gln3-Ure2 complex…

Response: Thank you for your suggestion. We have deleted the redundant part.

lane 465: Delete the “For the regulation of the gs and nr gene,” from the sentence.

Response: We have deleted accordingly.

lanes 467-469: “However, the regulation of these genes does not always depend on NmrA because nmrA silencing did not always inhibit their expression under the nitrate conditions.”

Please, be more specific. Avoid the phrase “does not always…”.

Response: We have re-written these sentences.

lane 470: What is NR in Aspergillus? Does NR refer to the nitrate reductase enzyme? There is no such historical abbreviation in Aspergilli, therefore, please avoid the usage of this abbreviation, or make it clear what does it mean. 

Response: Thank you for your suggestions. We have replaced NR with nitrate reductase enzyme.

lane 475: Basidiomycota

Response: We have revised the manuscript.

lanes 476-478: These sentences are redundant.

Response: We have deleted these sentences and re-written this part.

lanes 479-488: These sentences are redundant with the sentences in the Results section.

Response: We have deleted these sentences and re-written this part.

lanes 488-490: Redundant sentence.

Response: We have deleted these sentences.

lanes 490-499: These sentences are redundant with the Introduction.

Response: We have deleted these sentences and re-organized the Discussion part (Line 412-460).

We appreciate for your warm work earnestly, and hope that the correction will meet with approval.

Once again, thank you very much for your comments and suggestions.

Kind regards,

Jing Zhu

Round 2

Reviewer 2 Report (Previous Reviewer 1)

The author responded appropriately to my all comments. I recommend the following minor modifications.

Line 30, Correct "GA" to "ganoderic acid".

Line 60, Correct "areA or nmrA" to "AreA or NmrA".

Line 123, Table 1 is missing in the manuscript.

Line 164, Please note the composition of the extraction buffer. I think this extraction buffer is different from the extraction buffer containing 10% SDS described in line 137.

Line 190, I think that Ure2 interacts with Gln3 and Gat1, not AreA

Line 298, Correct "Western" to "western".

Line 386, Correct "the expression levels of hmgr and osc of nmrAi-3 and nmrAi-48 increased" to "the expression levels of hmgr and osc in nmrAi-3 and nmrAi-48 was increased".

Line 430, Correct "The expression of areA of WT increased" to "The expression of areA in WT was increased".

Line 438, Correct "The transcription factors NirA" to " The transcription factor NirA".

I feel that submitting to an English editing service will help the readers to better understand this manuscript.

Author Response

The author responded appropriately to my all comments. I recommend the following minor modifications.

 Response: Thank you for your suggestions. We have carefully revised the manuscript. The point-to-point replies are listed below.

Line 30, Correct "GA" to "ganoderic acid".

 Response: We have revised the manuscript.

Line 60, Correct "areA or nmrA" to "AreA or NmrA".

 Response: We have revised the manuscript.

Line 123, Table 1 is missing in the manuscript.

 Response: Table1 was added in page 16.

Line 164, Please note the composition of the extraction buffer. I think this extraction buffer is different from the extraction buffer containing 10% SDS described in line 137.

Response: Thank you for your suggestion. The extraction buffer for detecting enzyme activity is different from the extraction buffer for extracting protein. Therefore, we have added the composition of the extraction buffer in lines 162.

Line 190, I think that Ure2 interacts with Gln3 and Gat1, not AreA

 Response: Thank you for your suggestion. We have revised this description.

Line 298, Correct "Western" to "western".

 Response: We have revised accordingly.

Line 386, Correct "the expression levels of hmgr and osc of nmrAi-3 and nmrAi-48 increased" to "the expression levels of hmgr and osc in nmrAi-3 and nmrAi-48 was increased".

 Response: We have revised accordingly.

Line 430, Correct "The expression of areA of WT increased" to "The expression of areA in WT was increased".

 Response: We have revised accordingly.

Line 438, Correct "The transcription factors NirA" to " The transcription factor NirA".

 Response: We have revised accordingly.

I feel that submitting to an English editing service will help the readers to better understand this manuscript.

We have carefully revised the manuscript according to your and reviewer 2’s suggestions. The manuscript has been sent to MDPI office for English pre-edit services previously. The English editing ID is english-63239.

We appreciate for your warm work earnestly.

Thank you very much for your suggestions.

Kind regards,

Jing Zhu

Reviewer 3 Report (New Reviewer)

Review of version 2

The manuscript has been improved, although many grammatical errors remained in the text. I listed these errors below together with additional corrections.

line 30: In the abstract, write the whole name of GA, ganoderic acid.

line 52: by AreA activity

line 62: changed

line 71: affect the mycelial growth and the biosynthesis of active metabolites in G. lucidum.

line 77: The Ganoderma NmrA

line 106: RT-qPCR

line 115: RT-qPCR

line 140: 10 min

line 165: 12,000 g

line 168: The activity of nitrate reductase (NR) was ...

line 190: regulated

line 191: ... but the AreA-interacting proteins ....

lines 199-200: The NmrA protein contained an NMR-like Rossmann-type folded NAD+ and an NADP+ binding domain (Pfam05368, cd08947)(Fig. 1A).

line 202: The short chain dehydrogenase/reductase (SDR)

line 227: ... was used to verify the interaction of AreA with NmrA in ....

line 228: ... of the AreA contained ....

line 259: RT-qPCR

line 262: ...and nmrAi-48 was decreased ....

line 265: ....was decreased...

line 269: ...were cultured in ammonium or in nitrate as sole nitrogen sources.

line 270: ...silencing of nmrA ....     .....either in ammonium or nitrate.

line 273: ...was decreased...

line 276: ...was decreased...

line 227: ...silencing of nmrA ...

line 281> RT-qPCR

line 299: ...was decreased...

lines 300-301: At 6 h, approximately 80% of the AreA protein was still present, compared with the untreated strains.

line 302:....was decreased....

line 318:....was decreased....

line 322: ....was increased...

line 324: ....silencing of nmrA...

line 344: ....was increased...

line 345: ...was also increased...

line 352: ...was increased ...

line 354: ...silencing of nmrA...

line 372: ...was decreased ...

line 373: .....the diameters were decreased....

line 375: ...was decreased...

line 377: ...was decreased...

line 379: ...silencing of nmrA...

line 380: ...was increased...

line 385: ....synthase) were also investigated ...

line 386: ... was increased ...

line 388: ... was decreased ...

line 389: ...was increased ...

line 390: ... silencing of nmrA ...

line 411: was identified, and its interaction with AreA was shown in G. lucidum.

line 412: ...NmrA was shown to be required for the stability...

line 421: ....[19], the mutation .....                    (delete the and after the comma)

line 425: ....but also in nitrogen metabolism, ....

line 426: While the extracellular deaminases are not activated in the presence of preferred nitrogen source, they could be activated under nitrogen starvation in a wild type strain. In an nmr-1 mutant, however, these extracellular deaminases showed activation in the presence of preferred nitrogen source [47]. A significant increase of extracellular deaminases was observed in an nmrA deletion strain in the presence of preferred nitrogen source [REFERENCE is needed].

There is no nmrA deleted strain obtained in the referred literature numbered as 47.

line 430: ... was increased ...

line 432: ... silencing of nmrA ...

lines 437-438: The expression of these genes are not only induced by AreA.

lines 440-441: ....has a pivotal role in the expression of nitrogen regulated genes and nitrogen metabolism ...

lines 442-444: These studies not only demonstrate that other factors might participate in the regulation of the gene expression of nr and gs, but also reflect that the regulation of nitrogen utilization is a complex process.

line 445: ......and post-translational modifications can control the functioning of transcription factors, ....

line 450: ... was decreased ...

line 458: ...silencing of nmrA ...

line 461: Our studies show how the AreA activity is regulated in Basidiomycota.

Author Response

The manuscript has been improved, although many grammatical errors remained in the text. I listed these errors below together with additional corrections.

Response: Thank you for your suggestions. We have carefully revised the manuscript. The point-to-point replies are listed below.

line 30: In the abstract, write the whole name of GA, ganoderic acid.

Response: We have revised accordingly.

line 52: by AreA activity

Response: We have revised accordingly.

line 62: changed

Response: We have revised accordingly.

line 71: affect the mycelial growth and the biosynthesis of active metabolites in G. lucidum.

Response: We have revised accordingly.

line 77: The Ganoderma NmrA

Response: We have revised accordingly.

line 106: RT-qPCR

Response: We have revised accordingly.

line 115: RT-qPCR

Response: We have revised accordingly.

line 140: 10 min

Response: We have revised accordingly.

line 165: 12,000 g

Response: We have revised accordingly.

line 168: The activity of nitrate reductase (NR) was ...

Response: We have revised accordingly.

line 190: regulated

Response: We have revised accordingly.

line 191: ... but the AreA-interacting proteins ....

Response: We have revised accordingly.

lines 199-200: The NmrA protein contained an NMR-like Rossmann-type folded NAD+ and an NADP+ binding domain (Pfam05368, cd08947)(Fig. 1A).

Response: Thank you for your suggestions. We have added this information in the revised manuscript.

line 202: The short chain dehydrogenase/reductase (SDR)

Response: We have revised accordingly.

line 227: ... was used to verify the interaction of AreA with NmrA in ....

Response: We have revised accordingly.

line 228: ... of the AreA contained ....

Response: We have revised accordingly.

line 259: RT-qPCR

Response: We have revised accordingly.

line 262: ...and nmrAi-48 was decreased ....

Response: We have revised accordingly.

line 265: ....was decreased...

Response: We have revised accordingly.

line 269: ...were cultured in ammonium or in nitrate as sole nitrogen sources.

Response: We have revised accordingly.

line 270: ...silencing of nmrA ....     .....either in ammonium or nitrate.

Response: We have revised accordingly.

line 273: ...was decreased...

Response: We have revised accordingly.

line 276: ...was decreased...

Response: We have revised accordingly.

line 227: ...silencing of nmrA ...

Response: We have revised accordingly.

line 281> RT-qPCR

Response: We have revised accordingly.

line 299: ...was decreased...

Response: We have revised accordingly.

lines 300-301: At 6 h, approximately 80% of the AreA protein was still present, compared with the untreated strains.

Response: We have revised accordingly.

line 302:....was decreased....

Response: We have revised accordingly.

line 318:....was decreased....

Response: We have revised accordingly.

line 322: ....was increased...

Response: We have revised accordingly.

line 324: ....silencing of nmrA...

Response: We have revised accordingly.

line 344: ....was increased...

Response: We have revised accordingly.

line 345: ...was also increased...

Response: We have revised accordingly.

line 352: ...was increased ...

Response: We have revised accordingly.

line 354: ...silencing of nmrA...

Response: We have revised accordingly.

line 372: ...was decreased ...

Response: We have revised accordingly.

line 373: .....the diameters were decreased....

Response: We have revised accordingly.

line 375: ...was decreased...

Response: We have revised accordingly.

line 377: ...was decreased...

Response: We have revised accordingly.

line 379: ...silencing of nmrA...

Response: We have revised accordingly.

line 380: ...was increased...

Response: We have revised accordingly.

line 385: ....synthase) were also investigated ...

Response: We have revised accordingly.

line 386: ... was increased ...

Response: We have revised accordingly.

line 388: ... was decreased ...

Response: We have revised accordingly.

line 389: ...was increased ...

Response: We have revised accordingly.

line 390: ... silencing of nmrA ...

Response: We have revised accordingly.

line 411: was identified, and its interaction with AreA was shown in G. lucidum.

Response: We have revised accordingly.

line 412: ...NmrA was shown to be required for the stability...

Response: We have revised accordingly.

line 421: ....[19], the mutation .....                    (delete the and after the comma)

Response: We have revised accordingly.

line 425: ....but also in nitrogen metabolism, ....

Response: We have revised accordingly.

line 426: While the extracellular deaminases are not activated in the presence of preferred nitrogen source, they could be activated under nitrogen starvation in a wild type strain. In an nmr-1 mutant, however, these extracellular deaminases showed activation in the presence of preferred nitrogen source [47]. A significant increase of extracellular deaminases was observed in an nmrA deletion strain in the presence of preferred nitrogen source [REFERENCE is needed].

There is no nmrA deleted strain obtained in the referred literature numbered as 47.

Response: Sorry for the unclear description. In N. crassa, the homologous gene of nmrA is named nmr-1 according to the reference 47. We have revised the description. In N. crassa, glutamine reduced deaminase production. However, when nmr-1 (the nmrA homologous gene) was mutated, the enzyme activity of deaminase was increased under the glutamine condition

line 430: ... was increased ...

Response: We have revised accordingly.

line 432: ... silencing of nmrA ...

Response: We have revised accordingly.

lines 437-438: The expression of these genes are not only induced by AreA.

Response: We have revised accordingly.

lines 440-441: ....has a pivotal role in the expression of nitrogen regulated genes and nitrogen metabolism ...

Response: We have revised accordingly.

lines 442-444: These studies not only demonstrate that other factors might participate in the regulation of the gene expression of nr and gs, but also reflect that the regulation of nitrogen utilization is a complex process.

Response: We have revised accordingly.

line 445: ......and post-translational modifications can control the functioning of transcription factors, ....

Response: We have revised accordingly.

line 450: ... was decreased ...

Response: We have revised accordingly.

line 458: ...silencing of nmrA ...

Response: We have revised accordingly.

line 461: Our studies show how the AreA activity is regulated in Basidiomycota.

Response: We have revised accordingly.

We appreciate for your warm work earnestly.

Thank you very much for your suggestions.

Kind regards,

Jing Zhu

This manuscript is a resubmission of an earlier submission. The following is a list of the peer review reports and author responses from that submission.

Round 1

Reviewer 1 Report

The authors of this manuscript have been investigating the involvement of NmrA in regulation of AreA, a transcription factor regulating nitrogen metabolism, by silencing nmrA in the basidiomycete Ganoderma lucidum. The negative transcriptional regulator NmrA is known to be involved in the regulation of AreA in the ascomycetes, but its function had not been studied in basidiomycetes. Therefore, investigating the function of NmrA in G. lucidum is scientifically quite important. However, this manuscript has many concerns, as indicated below.

Major comments

1. The process for identification of the G. lucidum NmrA is not described. Did the authors use Blastp program? If so, what database was used and what sequence was entered as query? In addition, gene ID of G. lucidum nmrA gene should be provided.

2. In fig. 2A, the color of the agar medium for the negative control strains is obviously different from that for positive control and AreA+NmrA strains. All strains should be cultured on the same single agar medium.

3. In fig. 3A, the transcriptional expression level of nmrA in nmrA silencing strains was examined in CYM medium. This nmrA silencing efficiency also should be examined in the medium using ammonium or nitrate as nitrogen source since AreA protein level was examined in these media. This point is essential to accurately understand the effect of nmrA silencing on AreA protein levels.

4. In fig. 3B, the authors have shown the trimmed image in which only single band is found. However, in the original image of fig. 3, multiple bands have been detected. Importantly, the intensities of these bands are also strongly affected by nmrA silencing and nitrogen source. Therefore, these bands are considered not nonspecific, but may be posttranslational modified AreA. The authors need to examine this point carefully. In addition, no information is given about the antibodies used. Moreover, the band's position of the molecular weight standard should be indicated to allow the readers to consider the bands detected by western blotting.

5. In fig. 4A, AreA protein in nmrA silencing strain is suddenly disappeared 10 hours after CHX treatment. In the original image of fig. 4, the other proteins are also suddenly disappeared together with AreA. This is unnatural result. In addition, the graph showing the quantitation of the AreA/beta-tubulin protein level ratio is inappropriate because the different secondary antibodies were used for detection of AreA and beta-tubulin, and the exposure times for detection of these proteins should be different. The y-axis should be the percentage of AreA/beta-tubulin at each time with AreA/beta-tubulin at time 0 as 100%. The number of experiments performed is not shown, and all detectied images should be provided as original images. 

6. In fig. 7A, the resolution of provided images is too low. Also, because of the distorted shape of the colonies, please indicate in the images the area of measured length shown in fig. 7B.

7. Discussion is not sufficient. Why AreA accumulated into the nucleus in the nitrate medium and nmrA silencing has no effect on this? Why cytoplasmic AreA levels were significantly reduced by nmrA silencing? Why nmrA silencing increased gs gene expression level in ammonia medium and nr gene expression level in nitrate medium? Are these results consistent with previous reports in other microorganisms? Please discuss.

Other comments

1. The initial letter of the gene name in the Aspergillus species should be lowercased. I think the gene names in G. lucidum should also have the first letter lowercased consistent with other papers.

2. Line 60, As an abbreviation, "NCR" is first used here, so it needs to be written in full.

3. Line 92, Is the pH of the CMY medium adjusted?

4. Line 95, "plasmid transformation experiments" is an inappropriate term because the plasmid is not transformed.

5. Line 98, What does "sample" mean?

6. Lines 107-110, For the reader's understanding, the plasmid for nmrA silencing should be described in more detail. What promoter is used to express the antisense strand of nmrA? I could not find a plasmid named pAN7-ura3-dual-hyg in the reference 22.

7. Line 110, Correct "transformed" to "introduced".

8. Line 123-124, Correct sentence "pGBKT7-AreA and …" to "Yeast Y2H gold strain were co-transformed by pGBKT7-areA and pGADT7-nmrA".

9. Line 131, Correct "dots" to "colonies".

10. Lines 141 and 164, Correct "extract" to "extraction buffer".

11. Lines 172 and 173, As an abbreviation, "GA" is first used here, so it needs to be written in full.

12. Lines 202-206, Any given amino acid is usually represented by an "X", not a "#".

13. Lines 231-232, The interaction between AreA and NmrA was confirmed using yeast cells, not G. lucidum. This point should be clearly noted.

14. Line 298, I think "76%" is an incorrect.

15. Fig. 5B and 5D, Please indicate which is the criterion for the relative value.

16. Lines 321-322, The sentence "Whether the NmrA…" should be rewrote.

17. Line 322, Correct "detect" to "detected".

18. Lines 385-386 and 409, The term "nitrogen metabolism repression" has already been seen above (lines 43-44).

Reviewer 2 Report

Liu and colleges present their work on the identification of a homologue of the negative regulator of the GATA factor AreA that drives the regulation of the nitrogen-metabolite-repression circuit in most fungi.

Firstly, this manuscript requires extensive revision of the use of English. Some parts are difficult to understand and many sentences needs revision to clarify the content, data presentation and interpretation of results.

I also find a series of issues in the description of materials and methods and the description of experimental work and results.

Authors used ammonium sulfate as source of ammonium and sodium nitrate as for nitrate. They used 30mM NaNO3, and this renders a final concentration of 30mM NO3- anion, but the use of 30mM (NH4)2SO4 renders a final concentration of 60mM NH4+ cation. So authors are using in all experiments the double of ammonium concentration than nitrate, so they have stronger nitrogen repressible than inducible conditions. This mistake must be corrected all along the experimental work by performing the experiments at the correct concentration of ammonium.

It is not clear what authors consider as the homologue of nmrA gene in Ganoderma lucidum. They show 4 possible genes. Which is the correct one? To which gene is produced the antisense RNA? But also in relation with previous comments, how this antisense RNA is produced? It is required an experiment showing the expression levels of this antisense RNA. Which genes are subjected to qPCR control?  

It is well known that NmrA interacts with AreA, there is nothing new here. And would be also expected in G. lucidum if all domains are conserved. The two hybrid experiment is fine but simple and does not add nothing to what is known. Verification of this interaction by using biFC is unnecessary and in fact, those images showing this interaction are of very low quality; very low number of cells in controls and the verification of GFP reconstruction. Are domains conserved and the interacting regions? Authors could explore by Y2H interacting regions and show any specificity or conservation of mechanisms in Basidiomycetes.

Authors detect AreA using an unknown antibody in materials and methods. Blots are also largely trimmed and some bands with higher mobility have been deleted from figures. Authors measure the effect of antisense nmrA by looking to the AreA stability, but any effect on the expression levels of areA?

Figure 7A shows the phenotype of WT and NmrAi strains, obviously, the concentration of ammonium is not correct, but authors claim strong differences in the radial growth of these strains. This is not clearly visible when comparing panel A and panel B. It is a fact that NmrAi-3 and NmrAi-8 display differences in morphology in NH4 and NO3 plates. How authors explain this?